# *Allomyrina dichotoma* Larva Extract Ameliorates the Hepatic Insulin Resistance of High-Fat Diet-Induced Diabetic Mice

**DOI:** 10.3390/nu11071522

**Published:** 2019-07-04

**Authors:** Kyong Kim, Gong Deuk Bae, Minho Lee, Eun-Young Park, Dong Jae Baek, Chul Young Kim, Hee-Sook Jun, Yoon Sin Oh

**Affiliations:** 1Department of Food and Nutrition, Eulji University, Seongnam 13135, Korea; 2Lee Gil Ya Cancer and Diabetes Institute, Department of molecular medicine, Gachon University, Incheon 21999, Korea; 3Department of Food Technology and Services, Eulji University, Seongnam 13135, Korea; 4College of Pharmacy and Natural Medicine Research Institute, Mokpo National University, Jeonnam 58554, Korea; 5College of Pharmacy, Hanyang University, Ansan 15888, Korea; 6College of Pharmacy and Gachon Institute of Pharmaceutical Science, Gachon University, Incheon 21936, Korea

**Keywords:** *Allomyrina dichotoma* larva, hepatic insulin resistance, lipogenesis

## Abstract

*Allomyrina dichotoma* larva is a nutritional-worthy future food resource and it contributes to multiple pharmacological functions. However, its antidiabetic effect and molecular mechanisms are not yet fully understood. Therefore, we investigated the hypolipidemic effect of *A. dichotoma* larva extract (ADLE) in a high-fat diet (HFD)-induced C57BL/6J mice model. Glucose tolerance and insulin sensitivity in HFD-induced diabetic mice significantly improved after ADLE administration for six weeks. The levels of serum triglyceride (TG), aspartate aminotransferase (AST), alanine transferase (ALT) activity, and lipid accumulation were increased in the liver of HFD-fed mice, but the levels were significantly reduced by the ADLE treatment. Moreover, hepatic fibrosis and inflammatory gene expression in the liver from HFD-treated mice were ameliorated by the ADLE treatment. Dephosphorylation of AMP-activated protein kinase (AMPK) by palmitate was inhibited in the ADLE treated HepG2 cells, and subsequently reduced expression of lipogenic genes, such as *SREPBP*-*1c*, *ACC*, and *FAS* were observed. The reduced expression of lipogenic genes and an increased phosphorylation of AMPK was also observed in the liver from diabetic mice treated with ADLE. In conclusion, ADLE ameliorates hyperlipidemia through inhibition of hepatic lipogenesis via activating the AMPK signaling pathway. These findings suggest that ADLE and its constituent bioactive compounds are valuable to prevent or treat hepatic insulin resistance in type 2 diabetes.

## 1. Introduction

Nonalcoholic fatty liver disease (NAFLD), or fatty liver, is a chronic liver disease associated with hepatic dysfunction due to excessive triglyceride (TG) accumulation [1]. NAFLD is closely related to metabolic diseases, such as obesity, insulin resistance, hypertension, and dyslipidemia, and the incidence of NAFLD increases as obesity increases [2]. Abnormal dietary fat intake directly contributes to a casual role in hepatic lipid accumulation, such as TG, cholesterol, and lipid droplets in the cytoplasm of hepatocytes [3], and lipotoxicity that is caused by fat accumulation induces hepatic insulin resistance in the pathogenesis of type 2 diabetes [4]. 

The current drugs for treatment of NAFLD have considerable side effects, and their long-term treatment effect is not revealed and unsafe [5]. Therefore, an effective alternative to pharmacological therapy is needed, such as using natural products, which may be safe and beneficial in the treatment and prevention of NAFLD. 

Recently, according to the Food and Agriculture Organization of the United Nations, edible insects have been reported to be possible future human dietary supplements because they are a rich source of unsaturated fatty acids, proteins, minerals, vitamins, and other nutrients [6,7,8]. Various insects have already been used as traditional food or folk remedies in many countries, however, their consumption has been restricted by the lack of scientific proof of positive effects and safety to humans. *Allomyrina dichotoma*, also known as “Jangsupungdaeng-i” in Korea, is a species of rhinoceros beetle that is widely used in traditional medicine for its anti-hepatofibrotic, antineoplastic, and antioxidant effects [9,10,11,12]. The component of pharmacological activity was established in beetle larva [11]. However, whether *A. dichotoma* larva extract (ADLE) can prevent fatty liver or improve hyperglycemia in high-fat diet (HFD)-induced type 2 diabetes has not yet been investigated. Therefore, we investigated the effect of diabetic and liver steatosis of ADLE in diabetic mice induced by HFD-fed mice model, and its molecular mechanisms in palmitate-treated HepG2 cells.

## 2. Materials and Methods

### 2.1. Preparation of A. Dichotoma Larva Extract (ADLE)

*A. dichotoma* larva were purchased from Yechun Bugs land (Yecheon-gun, Gyungsangbuk-do, Korea). The dried *A. dichotoma* larva were powdered by using a grinder and extracted with ethanol/water (70:30, *v*/*v*) at a ratio of 20 mL/g for 3 h at 60 °C thrice, followed by filtration twice through filter paper through No. 2 filter paper (Advantec Toyo Roshi Kaisha, Ltd., Tokyo, Japan). The filtrate was concentrated in a vacuum rotary evaporator (*T <* 40 °C), lyophilized, and dissolved in deionized water. 

### 2.2. Cell Culture

The HepG2 cells, a human hepatoblastoma cell line, were cultured in Dulbecco’s modified Eagle’s medium (Gibco, Paisley, UK) containing 1% penicillin/streptomycin (Welgene, Daegu, Korea) and 10% fetal bovine serum (Gibco) in an atmosphere of 5% CO_2_ at 37 °C. 

### 2.3. Preparation of Palmitate and MTT Assay

Sodium palmitate (Sigma, St. Louis, MO, USA) was conjugated with 5% bovine serum albumin (Sigma) at a 1:3 volume ratio to make a 20 mM stock solution [13]. The HepG2 cells were exposed to 0.5 mM palmitic acid with or without ADLE for 24 h. Cell viability was determined by colorimetry using 3-(4,5-dimethylthiazolyl-2)-2,5-diphenyltetrazolium bromide (MTT, thiazolyl blue) (Duchefa Biochemie BV, Haarlem, the Netherlands). Insoluble formazan crystals were dissolved in 2-propanol and detected using a microplate spectrophotometer at 540 nm. 

### 2.4. Animals

Four-week-old C57BL/6J male mice were obtained from the Korea Research Institute Bioscience & Biotechnology (KRIBB, Daejeon, Korea). The mice were allowed free access to standard chow diet and water for a week. The mice were maintained at an ambient temperature (23 °C) with 12:12 h light–dark cycles with free access to food and water. All animal procedures were approved by the Institution Animal Care and Use Committee at Eulji University (EUIACUC-18-7). After a week of adaptation, the mice were fed a high-fat diet (HFD) of 60% fat (D12492; Research Diets, New Brunswick, NJ, USA) for 6 weeks (*n* = 33). Aged-matched control mice were fed a normal-fat diet (NFD) of 4.5% fat (Purina) for the same period (*n* = 6). Six weeks after the HFD, blood glucose levels and body weight were checked, and the mice with blood glucose levels >200–250 mg/dL were used for experiments. Diabetic mice were treated orally with ADLE (100 mg/kg/day) or vehicle (distilled water) (daily for 6 weeks) as described previously [14,15]. Metformin (Cayman Chemical, MI, USA) was orally administered (100 mg/kg/day) for 6 weeks as a positive control. Body weight and food intake were recorded weekly, and blood glucose was measured every 2 weeks for the 6 week feeding period. The food efficiency ratio (FER) was calculated based on the formula: FER = body weight increased during experimental period (g)/total food intake during the experimental period (g).

### 2.5. Intraperitoneal Glucose Tolerance Test and Intraperitoneal Insulin Tolerance Test

After fasting for 18 h, blood glucose was measured in whole blood from the tail vein (0 min) by using a one-touch glucometer (Johnson & Johnson, New Brunswick, NJ, USA). Subsequently, the glucose solution dissolved in phosphate-buffered saline (PBS) was injected intraperitoneally (2 g/kg), and blood glucose was detected at 30, 60, and 120 min. For the insulin tolerance test (ITT), the mice were fasted for 4 h, and blood glucose was measured in whole blood from the tail vein (0 min) by using a glucometer, the insulin solution dissolved in PBS was injected intraperitoneally (2 units/kg), and blood glucose was detected at 30, 60, and 120 min. An area under the curve (AUC) trapezoid model from 0 to 120 min after challenge was used to quantitatively evaluate glucose clearance activity. The AUC between any two time points was calculated as follows: (time difference in minutes between sequential reads) × (glucose level 1st time point + glucose level 2nd time point)/2) [16].

### 2.6. Biochemical Analysis in Blood

Serum lipid concentrations were determined with commercially available kits. The total cholesterol (TC), triglyceride (TG), and high-density cholesterol (HDL) levels in serum were measured based on the manufacturer’s instruction (Asan Pharmaceutical Co., Seoul, Korea). The concentration of low-density cholesterol (LDL) was calculated as (total cholesterol−HDL choles tero–[triglyceride/5]) based on the formula of Friedewald et al. [17]. Aspartate aminotransferase (AST) and alanine transferase (ALT), which are known to be a hepatic function marker, were measured based on the manufacturer’s instruction (Asan Pharmaceutical Co.). The units were expressed as IU/L of serum. All analyses were measured using a UV spectrometer (TECAN Group Ltd, Shanghai, China).

### 2.7. Assessments of Liver TG and TC

The method for measuring the contents of lipids in the liver was as described by Folch et al. [18]. Briefly, the liver (20–40 mg) was homogenized in cold PBS. A homogenate of 0.2 mL was extracted with methanol/chloroform (1:2) and centrifuged at 2500× *g* for 10 min. An aliquot of the organic phase was collected, dried under nitrogen, and resuspended in Triton X-100/ethanol mixture (1:1, *v*/*v*). The liver TG and TC levels were determined using its quantification kits (Asan Pharm.). Data were normalized for differences in protein concentration in the liver extracts.

### 2.8. Western Blot Analysis

The liver tissue and cell lysates were homogenized in mammalian protein extraction buffer (Sigma Chemical Co., St. Louis, MO. USA) in the presence of protease inhibitor cocktail (Sigma) and phenyl methane sulfonyl fluoride (Sigma, PMSF). The lysates were centrifuged at 12,000 rpm at 4 °C for 20 min. The protein contents of the supernatants were determined using protein assay dye reagent concentrate (Bio-Rad Laboratories, Hercules, CA, USA) based on the manufacturer’s instructions. The same concentration of protein was separated electrophoretically by sodium dodecyl sulfate-polyacrylamide gel electrophoresis and transferred to nitrocellulose blotting membranes (Amersharm, GE Healthcare Life Science, Germany). The antibodies used for western blotting included anti-SREBP-1 (1:1000; Abcam, Cambridge, UK), anti-ACC (1:1000; Cell Signaling Technology), anti-FAS (1:1000; Cell Signaling Technology), anti-AMPKα (1:1000; Cell Signaling Technology), anti-phospho-AMPKα (Thr172) (1:1000; Cell Signaling Technology), and anti-β-actin (1:2500; Abcam). The protein bands were visualized following an enhanced chemiluminescence method using an ELC kit (Millipore, USA). The bands were quantified using Quantity 1 version 4.6.7 software (Bio-Rad Laboratories).

### 2.9. Quantitative Real-Time Polymerase Chain Reaction

Total RNA was extracted from the mouse liver using Trizol reagent (Invitrogen, Grand Island, NY, USA) and synthesized with Primescript^TM^ 1st strand cDNA synthesis kit (Takara Bio Inc., Shiga, Japan) to prepare cDNA. Real-time polymerase chain reaction (PCR) was performed on the ABI real-time PCR system (Applied Biosystem Inc., Forster City, CA, USA) using SYBR Premix Ex Taq II, ROX plus (Takara Bio Inc., Shiga, Japan) based on the manufacturer’s instructions. Amplification was performed as follows: 10 min at 90 °C, 15 s at 95 °C, and 1 min at 60 °C for 40 cycles. Table 1 shows the gene-specific primers. Cyclophilin was used as a reference gene, and all results were normalized to the abundance of cyclophilin mRNA. The relative amounts of mRNAs were calculated using the 2^ΔΔCt^ method.

### 2.10. Biochemical Staining 

For Oil red O staining assay, parts of the liver were immediately fixed upon dissection in 4% buffered formaldehyde solution (pH = 7.4) for 24 h. Cryosections (10 μm) were incubated with Oil red O solution (Cayman Chemical, Ann Arbor, MI, USA) for 10 min. For Masson’s trichrome staining assay, paraffin embedded liver sections (4 μm), were fixed in acetone and deparaffinized, and stained for Masson’s trichrome (MT). The sections were detected using an Olympus DP70 digital camera (Olympus Co., Tokyo, Japan) by Olympus BX61 microscope (Olympus Co., Tokyo, Japan).

### 2.11. Statistical Analysis

The results were presented as the mean and standard deviations (SDs). All statistical analyses were performed using SPSS 20.0 software (IBM SPSS V20.0.0 for Windows, IBM Co., Armonk, NY, USA). Significant differences among the groups were analyzed using the LSD comparisons test. Statistical significance was set up at *p* < 0.05.

## 3. Results 

### 3.1. Changes of Body Weight, Food Intake, and Fasting Blood Glucose Level by HFD Administration in Mice

The HFD was administered to C57BL/6J mice for six weeks to induce type 2 diabetes. Changes in body weight, food intake, and blood glucose level were measured both in the initial and final points (six weeks) (Table 2). After six weeks, the body weights of the HFD group significantly increased by 2.3-fold as compared with the NFD group (*p* < 0.001), whereas food intake decreased in the HFD group. The fasting blood glucose levels also increased in the HFD group as compared with the NFD group.

### 3.2. Treatment of ADLE Reduced Body Weight and Blood Glucose Level of HFD-Induced Diabetic Mice

To investigate whether treatment of ADLE reduced diabetic phenotype, 100 mg/kg of ADLE was administered for six weeks. Metformin (Met, 100 mg/kg) was used as a positive control. During the period of administration, the body weights of the HFD group were significantly higher than those of the NFD group (*p* < 0.001), and the ADLE and Met treatment reduced the body weight as compared with the HFD group (Figure 1A). However, with regard to the changes in body weight in each group, the body weight of the Met treated with HFD group, but not in the ADLE group, significantly reduced as compared with that in the HFD group (Figure 1B). Food intake during the experimental period was decreased in the HFD group as compared with NFD group, but the significant differences were not observed in the HFD treated groups (Figure 1C). Moreover, the HFD-fed mice showed higher FER than those of the NFD-fed mice, and the FER was not changed by ADLE or the Met treatment in the HFD group (Figure 1D). Treatment with ADLE significantly decreased the fasting blood glucose level as compared with the HFD group at six weeks (*p* < 0.05), and the effect was similar with the Met-treated group (Figure 1E).

### 3.3. Treatment of ADLE Improved Glucose and Insulin Tolerance of HFD-Induced Diabetic Mice

The glucose tolerance test (GTT) and ITT at six weeks after HFD with or without ADLE were performed to investigate whether ADLE reduced glucose resistance and insulin insensitivity in diabetic mice. Figure 2A shows that glucose tolerance was impaired in HFD-induced mice as compared with NFD mice, and it was improved in ADLE-treated mice. The incremental AUC in the HFD group was higher than that in the NFD group (*p* < 0.001), and ADLE treatment significantly decreased (*p* < 0.05, Figure 2A,B). Increased insulin tolerance by HFD was reduced by the ADLE treatment, and a significant decrease in AUC of ITT value was observed as compared with HFD (*p* < 0.05, Figure 2C,D).

### 3.4. ADLE Altered Lipid Profiles, AST and ALT Levels in Serum of HFD-Induced Diabetic Mice

To determine the effect of ADLE on the serum lipid profile in diabetic mice, we compared the levels of total cholesterol (TC), triglyceride (TG), high-density lipoprotein (HDL), and low-density lipoprotein (LDL) in HFD with or without the ADLE treatment. Serum levels of TC, HDL, and LDL in the HFD group were significantly increased as compared with the NFD group (*p* < 0.01 or *p* < 0.001), but no significant differences were observed between the HFD and the ADLE-treated HFD group. However, increased levels of TG in the HFD group was significantly reduced by the ADLE treatment (*p* < 0.05), and the effect was similar to that of the Met-treated group (Figure 3A). Subsequently, the enzyme activity of AST and ALT was measured to determine the effect of ADLE on hepatic function. Figure 3B shows that the increased level of AST and ALT by HFD was significantly decreased with the ADLE and Met treatments. 

### 3.5. ADLE Reduced Hepatic Fibrosis and Expression of Inflammatory Genes in the Liver of HFD-Induced Diabetic Mice

Hepatic fibrosis and inflammation are important risk factor for the progression of NAFLD [19]. As ADLE showed anti-hepatofibrotic and anti-inflammatory activity [10,20], we checked whether ADLE can improve fibrosis and inflammation in the liver of HFD-fed mice. As shown in Figure 4A, hepatic steatosis demonstrating single large fat droplets in the cytoplasm of hepatocytes and ballooning of hepatocytes was clearly observed in the HFD-fed mice, but ADLE or Met treatment blocked the hepatic steatosis. Moreover perivenular/pericellular fibrosis and infiltration of inflammatory cells was detected in the HFD-fed mice and it was reduced by the ADLE and Met treatments (Figure 4A). The mRNA level of collagen, type 1, alpha 2 (Col1a2), which is one of the fibrotic genes, also increased in the liver lysate of HFD-fed mice and the level was reduced by ADLE-treated mice (*p* < 0.01, Figure 4B). Moreover, the ADLE treatment ameliorated the expression level of inflammatory genes such as monocyte chemoattractant protein 1 (MCP-1) and the tumor necrosis factor alpha (TNF-α) induced by HFD (*p* < 0.01 and *p* < 0.05, Figure 4C).

### 3.6. ADLE Reduced Lipid Accumulation in the Liver of HFD-Induced Diabetic Mice

We investigated hepatic lipid accumulation by Oil red O staining in liver sections because serum levels of TG, AST, and ALT were significantly reduced by the ADLE treatment in diabetic mice. Figure 5A shows that Oil red O stained cells in the liver (red spot) was increased in the HFD group as compared with those in the NFD group, and the ADLE and Met treatments remarkably decreased the stained cells (Figure 5A). Therefore, we checked the TG and TC levels in the liver lysates of each group. The TG and TC levels in the HFD group were significantly increased as compared with those in the NFD group, and the level was inhibited by the ADLE treatment (*p* < 0.05, Figure 5B).

### 3.7. ADLE Reduced Lipogenic Gene Expression in Palmitate-Treated HepG2 Cells

The HepG2 cells were treated with palmitate with or without ADLE to investigate the antilipogenic effect and molecular mechanisms of ADLE. When the cytotoxicity of ADLE in the HepG2 cells was checked, treatment with 0.1–0.5 mg/mL of extract for 24 h did not change the cell viability in contrast to a significant decrease in 1 mg/ml of extract treatment (Figure 6A). Therefore, 0.5 mg/mL ADLE was used. The HepG2 cells treated with palmitate have been commonly used to induce hepatic steatosis of NAFLD [21,22], and 500 μM palmitate was used [23]. When the intracellular TG level was observed, the ADLE ameliorated TG levels caused by 500 μM palmitate in HepG2 cells (*p* < 0.05, Figure 6B). The mRNA and protein levels of sterol regulatory element-binding protein (SREBP), fatty acid synthase (FAS) and acetyl-CoA carboxylase (ACC) were examined to determine whether lipogenic genes were involved in the decreased TG level by ADLE. Figure 6C shows that increased mRNA expressions of these genes by palmitate are decreased by the ADLE treatment. The protein expression levels were also upregulated by palmitate, and it was significantly reduced in the HepG2 cells following treatment with ADLE (Figure 6D). Because phosphorylation of AMP-activated protein kinase (AMPK) was reported to play a crucial role in hepatic lipid metabolism regulation [13], we examined the phosphorylation of AMPK in this condition. The basal expression of AMPK was not changed, but the phosphorylation of AMPK at Thr 172 (pAMPK) was decreased by palmitate, and it was ameliorated by the ADLE cotreatment (Figure 6E). Decreased pAMPK/AMPK ratio in the palmitate-treated HepG2 cells was also inhibited by the ADLE treatment (Figure 6E).

### 3.8. ADLE Reduced the Expression of Lipogenic Genes and Dephosphorylation of AMPK in the Liver of HFD-Induced Diabetic Mice

Because lipogenic genes and phosphorylation of AMPK were involved in the reduced lipid accumulation of the HepG2 cells treated with ADLE, this mechanism in the liver lysate of HFD-induced diabetic mice with or without ADLE was confirmed. Both mRNA and protein expression levels of SREBP, FAS, and ACC were increased in the liver of HFD, whereas, the NFD and ADLE treatments significantly inhibited these expressions (Figure 7A,B). Figure 7C shows that the pAMPK/AMPK ratios of the HFD group was significantly decreased as compared with those of the control group (*p* < 0.001), and it was recovered by the ADLE treatment (*p* < 0.01) (Figure 7C).

## 4. Discussion 

The liver is a vital organ that modulates various metabolic processes, including lipid metabolism and glucose homeostasis. Excessive intake of dietary fats leads to lipid accumulation in the liver and can cause type 2 diabetes, chronic diseases, and obesity. NAFLD is reportedly associated with insulin resistance during the development of type 2 diabetes [24,25], and these metabolic diseases must be controlled for a heathy life. Although several drugs used to treat NAFLD are currently available, satisfactory outcomes have not been achieved. Natural products have been considered as alternative treatments to prevent NAFLD via various mechanisms, such as antioxidant, anti-inflammation, and antidyslipidemic effects [26,27]. Recently, insects have gained attention as a source of an effective bioactive product in many countries, but scientific evidence regarding its safety, biological effects, and molecular mechanisms to be applied as a therapeutic agent is lacking.

Previous studies have shown that ADLE has antihepatofibrotic, antineoplastic, antibiotic, and antioxidant effects, but the antihepatotoxicity effect during the development of type 2 diabetes was not well reported. In this study, we clearly demonstrated that ADLE improved hyperlipidemia and hepatic function in the liver of HFD-fed NAFLD mouse model. 

Yoon et al. reported that the administration of ADL powder (3000 mg/kg/day) decreased HFD (35% carbohydrate and 45% fat)-induced body weight gain, and Kim et al. showed that intracerebroventricular cannulation of ADLE (1 μg) into 60% HFD-fed mice significantly reduced body weight by food intake regulation [15,28]. In our study, we did not observe food intake, FER, and body weight alteration, but treatment of HFD-fed mice with ADLE ameliorated insulin tolerance. The types of extract, treated concentration and/or duration, and mode of administration might have been responsible for the different effects of ADLE on body weight gain. However, these results suggested that the antidiabetic effect of ADLE was not due to decreased food intake or reduced body weight. Glucose tolerance of diabetic mice treated with ADLE was also significantly improved; the mechanism underlying insulin secretion or anti-apoptotic effects remains to be investigated.

Most of the circulating cholesterol in serum is found in three major lipoproteins such as very-low-density lipoproteins (VLDL), LDL, and HDL, as well, several studies have shown that increased serum TC contributes to the development of NAFLD [29]. Cholesterol is mainly synthesized in the liver, but hepatic cholesterol can be removed by its conversion to bile acids or by enhanced hepatobiliary secretion, as an essential step in the reverse cholesterol transport process [30]. In this study, we found that the serum TC levels were not changed by ADLE in contrast to the hepatic TC levels which were significantly decreased in the ADLE-treated HFD group. These results suggested that ADLE might have an effect on the enhanced clearance of cholesterol by increasing bile acid synthesis and excretion in liver. In this study, we did not check bile acid levels in HFD-fed mice and ADLE-treated HFD mice, further studies regarding ADLE and cholesterol catabolism to bile acids will be needed.

The serum lipid profile showed that TG, which is a hepatic insulin resistance marker [31], was significantly reduced in ADLE-treated HFD mice as compared with that in HFD mice. The serum HDL level was increased as a result of HFD and decreased by the ADLE treatment. This phenomenon may be accompanied by an increase in the TC level during prolonged HFD, and these results were consistent with those of other studies [32,33].

Elevated serum AST and ALT levels are commonly associated with hepatic steatosis and are used as a biomarker for hepatic injury [34]. In this study, HFD-fed mice showed significantly increased serum AST and ALT levels, but ADLE reversed this effect. Moreover, ADLE decreased HFD-induced hepatic fibrosis and inflammation. Lipotoxicity is a result of an imbalance between lipid uptake and utilization. The abnormal metabolism of FFAs and their derivatives is the main cause of intracellular lipotoxic injury [35] and the injured steatotic hepatocyte induces inflammation and fibrosis [36]. In our results, the numbers of lipid vacuoles, TG and TC levels in the liver were reduced, whereas, insulin sensitivity was increased in hyperlipidemic mice fed with ADLE. These results suggested that ADLE has a preventive effect on the development and progression of NAFLD, and the protective effect regarding hepatic lipotoxicity was due to reducing fibrosis and inflammation.

We also found that the accumulation of intracellular TG levels significantly increased in FFA-treated HepG2 cells, and the ADLE treatment of HepG2 cells exhibited an inhibitory effect on FFA-induced hepatic steatosis. At a molecular level, ADLE significantly downregulated the expression of SREBP-1c, FAS, and ACC in the HepG2 cells. SREBP-1c is a transcription factor, which is known to regulate de novo lipogenesis genes, such as ACC, FAS, and stearoyl-CoA desaturase (SCD)-1 [37,38,39]. We found that ADLE attenuated hepatic steatosis not only by altering the expression of genes related to lipogenesis but also by increasing the phosphorylation of AMPK. AMPK is known as a central regulator of multiple metabolic pathways, and its activation leads to the hepatic lipogenesis suppression and enhanced fatty acid oxidation [40,41]. Liver specific AMPK activation protects against diet-induced obesity and NAFLD [42]. Consistent with the in vitro results, the ADLE treatment downregulated the mRNA and protein level of lipogenesis genes and activated AMPK in the liver. Our results suggested that improvement in insulin sensitivity by ADLE is associated with the attenuation of fat storage in the liver via downregulation of lipogenesis genes and activation of AMPK.

We observed that antihyperlipidemia effects and AMPK activity by ADLE was similar to that by Met, which has been suggested as a potential drug for the treatment of NAFLD [43]. Therefore, our results suggested that ADLE is a potential therapeutic drug for decreasing hyperlipidemia and improving insulin sensitivity in patients with NAFLD and with type 2 diabetes. With regard to the administration of ADLE as a treatment for NAFLD and metabolic disease, quality analysis and its validation of the proper components of ADLE by fractionation is in progress. We found that some fractions showed an antioxidant effect, and therefore reduction of oxidative stress can be one of the mechanisms of amelioration of hepatic insulin resistance.

It has been reported that fat content is higher in the larval stage than in the adult stage, but the protein (Chitin) content was not different during the growth state [44,45]. Therefore, the bioactive component changes during the growth stage and is dependent on the nutritional source. Moreover, the nutritional environment including the type of food, health status, and parental size will have an affect, and therefore further studies regarding the amounts of bioactive component in the insects during development will be needed. Moreover, for application of the effect of ADLE on antihepatotoxicity and antihyperglycemia in the human subject, many studies such as toxicity, allergies, and production should be investigated.

## 5. Conclusions

ADLE improved hepatic insulin resistance by preventing lipid accumulation in HFD-fed mice and human hepatocytes by inhibiting FAS, ACC, and SREBP-1c and activating AMPK phosphorylation in the liver tissue. These results support the potential therapeutic role of ADLE in type 2 diabetes by preventing HFD-induced hepatic insulin resistance. 

## Figures and Tables

**Figure 1 nutrients-11-01522-f001:**
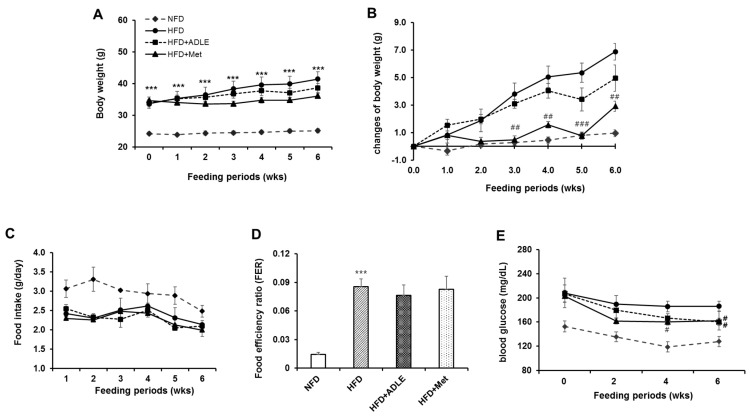
Treatment of ADLE reduced body weight and blood glucose level of HFD-induced diabetic mice. High-fat diet-induced diabetic mice were treated with vehicle, *Allomyrina dichotoma* larva extract (ADLE, 100 mg/kg) or metformin (Met, 100 mg/kg) for 6 weeks. (**A**) Body weight for 6 weeks, (**B**) changes in body weight for 6 weeks, (**C**) food intake (g/day), (**D**) FER over 6 weeks, and (**E**) fasting blood glucose levels for 6 weeks. Values are presented as means ± SD (*n* = 6–8). *** *p* < 0.001 vs. NFD group. ^#^
*p* < 0.05, ^##^
*p* < 0.01, and ^###^
*p* < 0.001 vs. HFD group. NFD, normal-fat diet; HFD, high-fat diet; HFD + ADLE, high-fat diet + ADLE; HFD + Met, high-fat diet + Met.

**Figure 2 nutrients-11-01522-f002:**
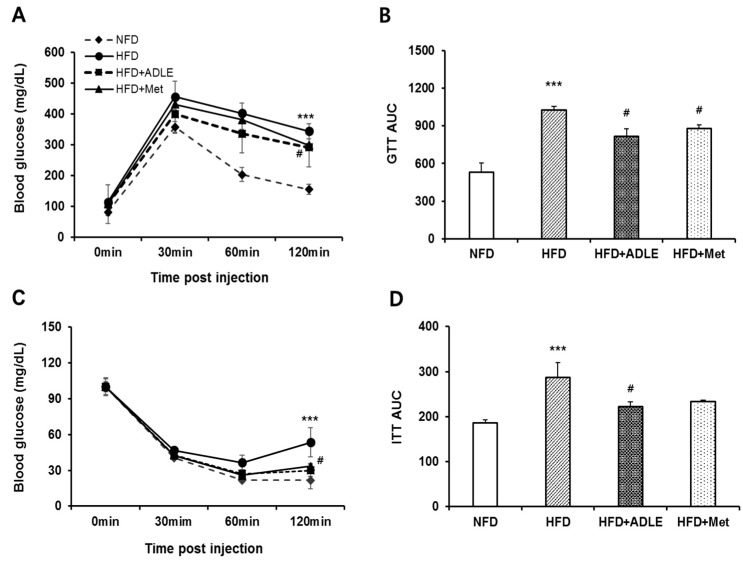
Treatment of ADLE improved glucose and insulin tolerance of HFD-induced diabetic mice. Mice were treated as described in Figure 1. (**A**) Intraperitoneal glucose tolerance test (GTT), (**B**) area under the curve (AUC) of GTT, (**C**) intraperitoneal insulin tolerance test (ITT), and (**D**) area under the curve (AUC) of ITT. Values are presented as mean ± SD (*n* = 6–8). *** *p* < 0.001 vs. NFD group. ^#^
*p* < 0.05 vs. HFD group. NFD, normal-fat diet; HFD, high-fat diet; HFD + ADLE, high-fat diet + ADLE; HFD + Met, high-fat diet + Met.

**Figure 3 nutrients-11-01522-f003:**
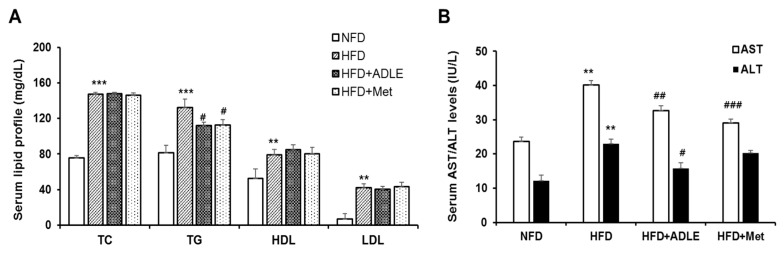
ADLE altered lipid profiles, AST, and ALT levels in serum of HFD-induced diabetic mice. Mice were treated as described in Figure 1. After 6 weeks of ADLE treatment, serum was collected, and (**A**) lipid profiles: total cholesterol (TC), triglyceride (TG), high-density lipoprotein (HDL), and low-density lipoprotein (LDL) and (**B**) aspartate aminotransferase (AST) and alanine aminotransferase (ALT) were measured by specific assay kits. The results are expressed as means ± SD (*n* = 6–8). ** *p* < 0.01, *** *p* < 0.001 vs. NFD group. ^#^
*p* < 0.05, ^##^
*p* < 0.01, and ^###^
*p* < 0.001 vs. HFD group. NFD, normal-fat diet; HFD, high-fat diet; HFD + ADLE, high-fat diet + ADLE; HFD + Met, high-fat diet + Met.

**Figure 4 nutrients-11-01522-f004:**
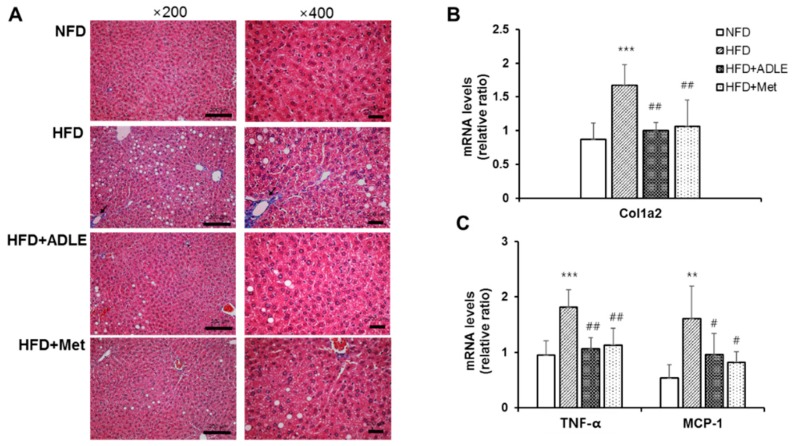
ADLE reduced hepatic fibrosis and inflammation in the liver of HFD-induced diabetic mice. (**A**) Liver sections were stained with Masson’s trichrome (scale bar = 200 μm) and (**B**,**C**) the mRNA expression of Col1a2, TNF-α, and MCP-1 in the liver was determined by RT-PCR. The results are expressed as means ± SD (*n* = 6–8). ** *p* < 0.01, *** *p* < 0.001 vs. NFD group. ^#^
*p* < 0.05 and ^##^
*p* < 0.01 vs. HFD group. NFD, normal-fat diet; HFD, high-fat diet; HFD + ADLE, high-fat diet + ADLE; HFD + Met, high-fat diet + Met.

**Figure 5 nutrients-11-01522-f005:**
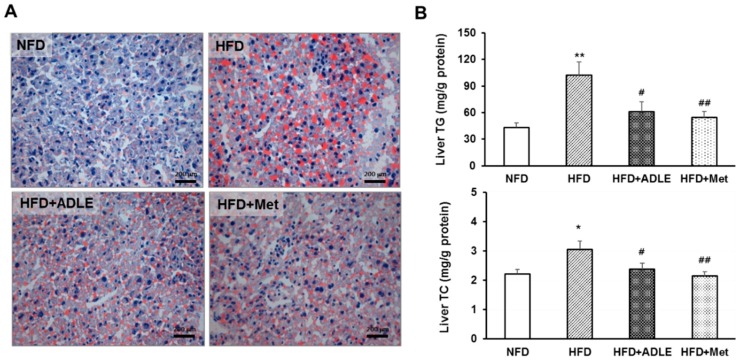
ADLE reduced lipid accumulation in the liver of HFD-induced diabetic mice. (**A**) Mice were treated as described in Figure 1. After 6 weeks of ADLE treatment, sections of liver tissue were stained with Oil red O staining (scale bar = 200 μm). (**B**) Hepatic triglyceride (TG, mg/g protein) and total cholesterol (TC, mg/g protein) levels from liver lysates were measured by assay kits. The results are expressed as means ± SD (*n* = 6–8). * *p* < 0.05 and ** *p* < 0.01 vs. NFD group. ^#^
*p* < 0.05 and ^##^
*p* < 0.01 vs. HFD group. NFD, normal-fat diet; HFD, high-fat die; HFD + ADLE, high-fat diet + ADLE; HFD + Met, high-fat diet + Met.

**Figure 6 nutrients-11-01522-f006:**
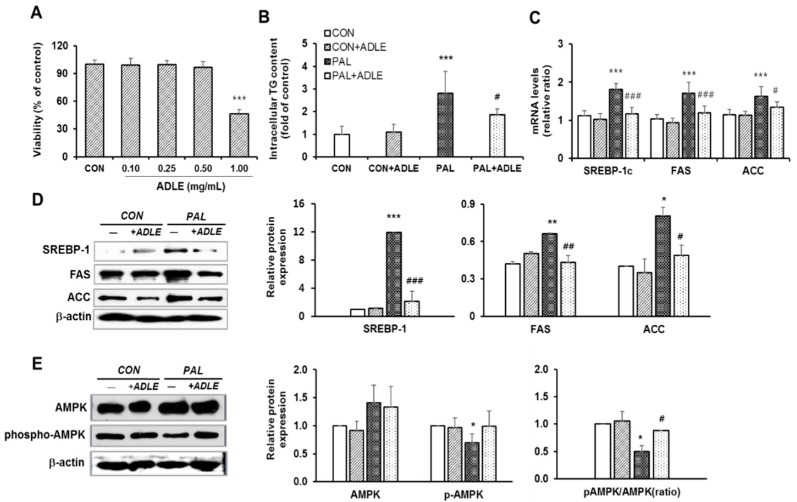
ADLE reduced lipogenic gene expression in palmitate-treated HepG2 cells. (**A**) HepG2 cells were cultured with various concentrations of ADLE for 24 h and cell viability was determined by MTT assay. (**B**) HepG2 cells were treated with 0.5 mM palmitate (PAL) with or without ADLE (0.5 mg/ml) for 24 h and the intracellular TG levels were measured by triglyceride assay kit. (**C**) The cells were treated as described in Figure 6B, and mRNA levels (SREBP-1c, FAS, and ACC) were determined by qRT-PCR analysis. The mRNA levels were normalized with those of cyclophilin. (**D**) Protein expression levels of SREBP-1, FAS, and ACC were measured by western blot analysis. (**E**) Protein expression levels of AMPK and phospho-AMPK were measured by western blot analysis. The intensity of each band was measured with Quantity one software, and the relative quantity was calculated over β-actin. The results are expressed as the mean ± SD (*n* = 3–5). * *p* < 0.05, ** *p* < 0.001, and *** *p* < 0.001 vs. untreated control (CON). ^#^
*p* < 0.05, ^##^
*p* < 0.001, and ^###^
*p* < 0.001 vs. 0.5 mM PAL only.

**Figure 7 nutrients-11-01522-f007:**
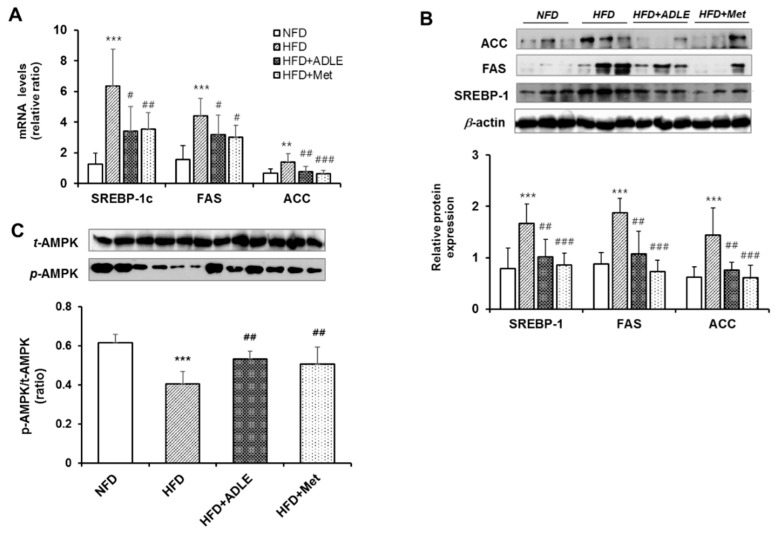
ADLE reduced the expression of lipogenic genes and dephosphorylation of AMPK in the liver of HFD-induced diabetic mice. Mice were treated as described in Figure 1. After 6 weeks of ADLE treatment, total RNA and protein were extracted from the liver as described in the Materials and Methods section. (**A**) The qRT-PCR analysis of gene expression for SREBP-1c, FAS, and ACC was normalized with those of cyclophilin. (**B**) Protein expression levels of SREBP-1, FAS, and ACC were measured by western blot analysis. The intensity of the bands was quantified by densitometric analysis and normalized with corresponding β-actin. (**C**) Western blots of phosphorylated AMPK (pAMPK) in the liver were performed, and the intensity of the bands was quantified by densitometric analysis and normalized with AMPK. Results are calculated as the means ± SD (*n* = 6–8). ** *p* < 0.01 and *** *p* < 0.001 vs. NFD group. ^#^
*p* < 0.05, ^##^
*p* < 0.01, and ^###^
*p* < 0.001 vs. HFD group. NFD, normal-fat diet; HFD, high-fat diet; HFD + ADLE, high-fat diet + ADLE; HFD + Met, high-fat diet + Met.

**Table 1 nutrients-11-01522-t001:** Primer sequences for real-time PCR.

Gene	Forward (5′-3′)	Reverse (5′-3′)
SREBP-1c	5′-CTTCTGGAGACATCGCAAAC-3′	5′-GGTAGACAACAGCCGCATC-3′
ACC	5′-AGGAAGATGGCGTCCGCTCTG-3′	5′-GGTGAGATGTGCTGGGTCAT-3′
FAS	5′-CTTGGGTGCTGACTACAACC-3′	5’-GCCCTCCCGTACACTCACTC-3′
Col1a2	5’-CCGTGCTTCTCAGAACATCA-3’	5’-CTTGCCCCATTCATTTGTCT-3’
MCP-1	5’-GCAGTTAACGCCCCACTCA-3’	5’-CCAGCCTACTCATTGGGATCA-3’
TNF-α	5’-CCAACGGCATGGATCTCAAAGACA-3’	5’-AGATAGCAAATCGGCTGACGGTGT-3’
Cyclophilin	5′-TGGAGAGCACCAAGACAGACA-3′	5′-TGCCGGAGTCGACAATGAT-3′

**Table 2 nutrients-11-01522-t002:** Changes in body weight gain, food intake and blood glucose in C57BL/6J mice for 6 weeks of on Type 2 diabetes induction.

Groups	Body Weight (g)		Gained Body Weight (g)	Food Intake (g/Day)	Blood Glucose (mg/dL)
	Initial	Final			
NFD (*n* = 6)	16.32 ± 1.28	24.17 ± 0.44	7.85 ± 0.48	3.09 ± 0.26	152.50 ± 9.12
HFD (*n* = 33)	16.11 ± 1.71	34.15 ± 1.39 ***	18.04 ± 0.61 ***	2.44 ± 0.16 ***	208.28 ± 8.30 ***

Values are mean ± SD. NFD, normal-fat diet; HFD, high-fat diet. *** Significant difference between group at *p* < 0.001 by *t*-test.

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
