# Peer review of "Allomyrina dichotoma Larva Extract Ameliorates the Hepatic Insulin Resistance of High-Fat Diet-Induced Diabetic Mice"

_nutrients, 2019, doi:10.3390/nu11071522_

Round 1
Reviewer 1 Report
Reviewer’s comment
This study primarily focuses on the effects of allomyrina dichotoma larva extract (ADLE) in the diabetic mice induced by high-fat diet. Several beneficial effects of the extract were obtained from this study. However, the interpretations and analyses on the results are inappropriate or insufficient in this study. Some revisions are required for the better quality of this article.
Major
#1.The authors should evaluate the hepatic histology, including activity, fibrosis and steatosis, in the experimental animals at 6 weeks. It is extremely important to confirm whether the liver in the animal might progress to nonalcoholic steatohepatitis or not at this point.
#2. A previous study elucidated the beneficial effects of ADLE on hepatic fibrosis and inflammation. Therefore, the anti-fibrotic and anti-inflammatory effects of the extract should be investigated in the experimental animals.
#3. The preliminary study to determine the administration dose of ADLE in vitro study was shown in Fig.5A. Likewise, the authors should perform the preliminary studies on the optimal administration doses of ADLE and Met in the experimental animals.
#4. Alanine aminotransferase (ALT) should be monitored for the evaluation of hepatic inflammation instead of AST (GOT). The name of GOT should be avoided, because the name was changed to AST.
#5. The food intake volume in each group should be noted during experimental period in order to calculate the FER. As shown in Fig.1C, the FER in ADLE group or MET group was almost equivalent to that in HFD group. The authors should describe what the result means.
#6. A previous report revealed the components of ADLE. The authors should speculate which component was the most responsible for the beneficial effects.
Minor
#1. The parameter of Y-axis in Fig. 3B is missing.
#2. ADL power should be corrected to “ADLE powder” (page 11, line 305).
#3. There are some unnatural wording in the text. The article should be checked by a native speaker. (ex, reduced body weight gain (page 11, line 306)).
#4. Letter sizes in line 239 and 259 (page 9) are different from those in the other lines.
Author Response
* Major comments
1) Response: We fully understand the reviewer’s comments. The hepatic steatosis develops after 6-12 weeks of HFD-fed mice (Lau JK et al., J Pathol, 2017; Gautjier MS et al., British J Nutrition, 2006). Although we did not check the NAFLD at the beginning of an experiment, symptoms of high glucose levels and insulin insensitivity is closely related with NAFLD (Nakamura A et al., J Diabetes Invest, 2011). When we check the hepatic fibrosis using Masson’s trichorome staining after 6 weeks ADLE treatment, collagen deposition was increased in the liver from HFD-fed mice compared with NFD-fed mice and it was significantly reduced by ADLE treatment. These results suggested that ADLE has preventive effect of development and progression of NAFLD. We added this results in the revised Fig. 4 and added information in the revised manuscript (page 13, line 365).
2) Response: Thank you for your valuable comment. To investigate whether anti-fibrotic and anti-inflammatory effect of ADLE, we checked mRNA expression level of fibrosis related gene (col1a2) and inflammatory genes (MCP-1 and TNF-α) in the liver of experimental group. We observed that col1a2 mRNA level was increased in the liver lysate of HFD-fed mice and it was significantly decreased by ADLE treatment. Moreover, ADLE treatment ameliorated the expression level of inflammatory genes such as MCP-1 and TNF induced by HFD. These results suggested that ADLE showed beneficial effect on hepatic fibrosis and inflammation as previously reported (Lee JH et al., J Micrbiol. Biotechnol., 2019; Yoshilawa K et al., FEBS Lett, 1999). We added this results and discussed in the revised manuscript (page 8, line 230)
3) Response: A literature review was performed and found that ADLE was used at a concentration of 100 ~ 3000 mg/kg into orally in diabetic mice (Yoon YI et al, Nutrients 2015; Im AR et al., Nutrients, 2019, ref. 14, 15 in the revised manuscript). Previous reports suggested that 100 mg/kg was effective, therefore we used this dose in our study. Moreover, same dose of Met (100mg/kg) was used as compared with the effect of ADLE.
4) As the reviewer’s comment, we investigated ALT activity in the liver from each groups. As shown in revised Fig 3B, serum ALT in the HFD-fed mice was significantly increased compared with NFD-fed mice and it was decreased in ADLE treatment. These results confirmed that ADLE exhibits hepatoprotective effects in HFD diabetic mice. We added this results in the revised manuscript (page 7, line 219, 220). As the reviewer’s comments, we changed from GOT to AST in the revised manuscript (page 7, line 211).
5) Response : As the reviewer’s comment, we exhibited food intake volume during experimental period as revised Fig.1C. We found that all the HFD groups showed low amount of food intake compared with NFD group probably due to a high energy density and there was no significant difference among the HFD groups. Therefore, HFD groups showed higher food efficiency ratio (FER) than those showed by the NFD group, but FER was not changed by ADLE or Met treatment in HFD group. These results suggested that anti-diabetic effect of ADLE was not due to decreased food intake or reduced body weight. We added this information in the revised manuscript (page 5, line 180; page 13, line 226, 340).
6) Response : As the response to reviewer’s comment, we are trying to validate the ingredient contained in the ADLE by qualitative analysis and found that some fractions showed anti-oxidant effect. We are prepared subsequent manuscript regarding component analysis of ADLE and its anti-diabetic function based on the anti-oxidant effect. Although we could not describe active ingredient of ADLE in this manuscript, but reduction of oxidative stress can be one of the mechanisms of amelioration of hepatic insulin resistance. As the reviewer’s suggestion, we added this information in the discussion section of revised manuscript (page 14, line 386).
* Minor comments
1) Response: We added Y-axis in the revised Fig 3B. (page 8, line 222).
2) Response: As Yoon et al. used ADL powder not ADL extract, we changed to ‘ADL powder’ in the revised manuscript (page 13, line 333).
3) Response: The manuscript has been reviewed and edited by a professional English editing service to correct linguistic errors and improve the clarity by revising confusing sentences.
4) Response: We set font size the same in the revised manuscript.
Reviewer 2 Report
This study was designed to examine the effects of Allomyrina dichotoma larva extract (ADLE) on hepatic steatosis and insulin resistance in diabetic mice induced by HFD and in palmitate-treated HepG2 cells. The data showed that ADLE improved insulin resistance, and that ADLE reduced the lipogenic genes and dephosphorylation of AMPK. The study is well conducted and the results seem to be of interest. However, the reviewer has the following concerns:
Major concerns:
1. Line 59; The authors should indicate the ingredient contained in the ADLE by qualitative analysis. Additionally, the authors should discuss the function of ADLE in the vision of potential active ingredient.
2. In Figures 3A and 4B; The authors demonstrate that no significant differences in the serum TC levels were observed between the HFD and ADLE-treated HFD group. However, the TC levels in the livers of HFD group were significantly inhibited by ADLE treatment. The authors should provide the discussion for the TC metabolisms in the serum and livers.
3. In the Graphical abstract; there was no information of hepatic insulin resistance.
Minor points:
1. Line 84; Please indicate the percent of fat of the normal-fat diet (NFD).
2. Line 87; The authors should provide the explanation for the selection of a dose of the 100 mg/kg ADLE in this study. Additionally, will this dose be achievable in humans?
3. Line 166; How many animals were there in each group (NFD and HFD) in Table 2?
4. In Figure 4A; Please insert a scale bar.
5. In Figures 5 and 6; The authors confuse SREBP-1 and SREBP-1c. According to the Materials and methods, the mRNA level of SREBP-1c was analyzed by quantitative real-time PCR. Please check the labels of SREBP-1 and SREBP-1c.
6. Will the ingredient contained in A. dichotoma larva change in the growth stage?
Author Response
* Major points
1) Response: Thanks for your valuable comment. We are trying to validate the ingredient contained in the ADLE by qualitative analysis and found that some fractions showed anti-oxidant effect. We are prepared subsequent manuscript regarding component analysis of ADLE and its anti-diabetic function based on the anti-oxidant effect. Although we could not describe active ingredient of ADLE in this manuscript, but reduction of oxidative stress can be one of the mechanisms of amelioration of hepatic insulin resistance. As the reviewer’s suggestion, we added this information in the discussion section of revised manuscript (page 14, line 386).
2) Response: Most of circulating cholesterol in serum is found in three major lipoproteins such as very low density lipoprotein (VLDL), low density lipoprotein (LDL) and high density lipoprotein (HDL) and several studies have shown that increased serum total cholesterol (TC) contributes to the development of NAFLD (Walenbergh SM et al., Expert Rev Gastroenterol Hepatol. 2015). Although cholesterol was mainly synthesized in liver, hepatic cholesterol can be removed by its conversion to bile acids or by enhanced hepatobiliary secretion, as essential step in the reverse cholesterol transport process (Arguello G et al., BBA-Mol Basis of Disease, 2015). As the reviewer’s comment, serum TC levels were not changes by ADLE treatment in contrast to hepatic TC levels was significantly decreased in ADLE-treated HFD group. These results suggested that ADLE might be affect enhance clearance of cholesterol by increasing bile acid synthesis and excretion in liver. In this study, we did not check bile acid levels in HFD-fed mice and ADLE treated HFD-fed mice, further studies regarding ADLE and cholesterol catabolism to bile acids will be needed. We added this information in the discussion section of revised manuscript. (page 13, line 343).
3) Response: As reviewer’s comment, we added information of hepatic insulin resistance and revised graphical abstract in the revised manuscript (page 2, line 33).
* Minor points
1)Response : We used 4.5% fat of the NFD and added this information in the revised manuscript (page 3, line 85)
2) Response: A literature review was performed and found that ADLE was used at a concentration of 100 ~ 3000 mg/kg into orally in diabetic mice (Yoon YI et al, Nutrients 2015; Im AR et al., Nutrients, 2019). Previous reports suggested that 100 mg/kg was effective (ref 14, 15 in the revised manuscript), therefore we used this dose in our study. When we adjust doses from animal studies (100 mg/kg) to human studies (dose equivalent to humans), 8.1 mg/kg of ADLE is appropriate. However, for application the effect of ALDE on anti-hepatotoxicity and anti-hyperglycemia in the human subject, many studies such as toxicity, allergies and production will be needed in the future. We added this information in the revised manuscript (page 14, line 395).
3)Response: We used 6 and 33 mice in NFD group and HFD group, respectively. We added this information in the revised Table 2 (page 5, line 168).
4) Response: We inserted a scale bar in the revised Fig. 5A (page 10, line 258).
5)Response: We fully understand the reviewer’s comment. We checked the labels of SREBP-1 and SREBP-1c and corrected in the revised figure and manuscript (page 11, line 292; page 12, line 310).
6) Response: Thanks for your valuable comment. It was reported that fat content is higher in the larval stage than in the adult stage, but the protein (Chitin) content was not different during the growth state (Elhassan M et al., Foods, 2019; Sin CS et al., Int J Biol Macro, 2019). Therefore, bioactive component changes during the growth stage is dependent on the nutritional source. Moreover, nutritional environment including type of food, health status, and parental size will be affect, further studies regarding amounts of bioactive component in the insects during development will be needed. We added this information in the revised manuscript (page 14, line 390).
Round 2
Reviewer 1 Report
The revised article seems to be markedly improved.
However, I found the paradoxical results in this study.
Oil red-stained cells were increased in HFD-mice (Fig.5A)
But, hepatic steatosis was not observed in the liver of HFD-mice (Fig.4A).
The balooning of hepatocytes was also absent in HFD-mice.
The picture of the liver in HFD-mice was not clear (Fig.4A)..
The author should demonstrate the typical pericellular and/or perivenular fibrosis and infiltration of inflammatory cells in those mice.
Author Response
Response : We fully understand the reviewer’s comment. To investigate whether exacerbation of hepatic steatosis in HFD-fed mice, repeat analysis using another sections was performed. As shown in revised figure 4A, hepatic steatosis demonstrating single large fat droplets in the cytoplasm of hepatocytes and ballooning of hepatocytes was clearly observed in the HFD-fed mice. Moreover, the results showed that ADLE or Met treatment blocked the hepatic steatosis. As the reviewer’s comment, we demonstrated the perivenular/ pericellular fibrosis and infiltration of inflammatory cells in the HFD-fed mice with or without ADLE and/or Met treatment. We added this information in the revised manuscript (Fig 4A, page 8, line 235; page 13, line 365, 369).
Reviewer 2 Report
This version of the manuscript has been improved. No further comments.
Author Response
Thanks for review our manuscript.